# Effect of an Eleven-Day Altitude Training Program on Aerobic and Anaerobic Performance in Adolescent Runners

**DOI:** 10.3390/medicina56040184

**Published:** 2020-04-16

**Authors:** Petr Bahenský, Václav Bunc, Pavel Tlustý, Gregory J. Grosicki

**Affiliations:** 1Department of Sports Studies, Faculty of Education, University of South Bohemia, 371 15 České Budějovice, Czech Republic; 2Physical Training and Education, Sports Motor Skills Laboratory, Faculty of Sports, Charles University, 165 52 Prague, Czech Republic; Bunc@ftvs.cuni.cz; 3Department of Mathematics, Faculty of Education, University of South Bohemia, 371 15 České Budějovice, Czech Republic; tlusty@pf.jcu.cz; 4Biodynamics and Human Performance Center, Georgia Southern University, Savannah, GA 31419, USA; grosicki@georgiasouthern.edu

**Keywords:** endurance, training camp, testing, running, youth, VO_2_max, lactate tests

## Abstract

*Background and Objectives:* We evaluated the effect of an eleven-day altitude training camp on aerobic and anaerobic fitness in trained adolescent runners. *Materials and Methods:* Twenty adolescent (14–18 yrs) middle- and long-distance runners (11 males and 9 females; 16.7 ± 0.8 yrs), with at least two years of self-reported consistent run training, participated in this study. Eight of the subjects (4 females/4 males) constituted the control group, whereas twelve subjects (5 females/7 males) took part in a structured eleven-day altitude training camp, and training load was matched between groups. Primary variables of interest included changes in aerobic (VO_2_max) and anaerobic (30 s Wingate test) power. We also explored the relationships between running velocity and blood lactate levels before and after the altitude training camp. *Results:* Following 11 days of altitude training, desirable changes (*p* < 0.01) in VO_2_max (+13.6%), peak relative work rate (+9.6%), and running velocity at various blood lactate concentrations (+5.9%–9.6%) were observed. Meanwhile, changes in Wingate anaerobic power (+5.1%) were statistically insignificant (*p* > 0.05). *Conclusions:* Short duration altitude appears to yield meaningful improvements in aerobic but not anaerobic power in trained adolescent endurance runners.

## 1. Introduction

Endurance training yields morphological and functional adaptations of the human organism [1]. Concomitant adaptations of numerous physiological systems (e.g., cardiovascular, musculoskeletal, etc.) results in an increase in oxygen consumption (VO_2_max) and improvements in power output [1,2]. Coalescence of these adaptations yields superior endurance-related performance outcomes, reflected by improvements in running performance. During maturation, in adolescent age (14–18 years) [3], functional and morphological systems are developing. Data dealing with the effects of endurance training on the development of physiological systems in adolescent runners are sparse [4].

In adults, altitude training provides a unique stimulus to potentiate many of the physiological benefits seen with endurance training [5,6,7,8]. More specifically, the available evidence confirms the development of cardiorespiratory fitness with altitude training [9]. However, fewer studies have sought to examine the influence of altitude training in adolescent individuals. Buchheit et al. [5] provided evidence that high intensity interval training in a hypoxic environment may not be optimal for cardiorespiratory and neuromuscular development in adolescent runners. Meanwhile, findings from our group [10] and Saltin et al. [7] provide evidence for a positive benefit of altitude training on cardiorespiratory fitness parameters in young runners. These contrasting findings highlight the uncertain benefits of altitude training for adolescent endurance athletes, a unique population with highly dynamic and evolving physiological systems.

Endurance training-mediated changes in cardiorespiratory fitness level are most frequently quantified via VO_2_max [11,12,13] and anaerobic threshold [14,15,16,17]. Changes in anaerobic performance following acute altitude exposure, most frequently quantified via evaluation of 30-s Wingate anaerobic power, are less studied [18]. Since there is no doubt that training at higher altitudes is a potent stimulus to achieve peak performance in adults runners [6,8,10,11], and because acclimatization and training become more effective with every stay at higher altitude [19], it is interesting to see how young runners respond to these conditions. In adulthood, acclimatization to a higher altitude can be quicker, and training there can be more efficient. The overarching goal of the present study was to characterize aerobic (VO_2_max) and anaerobic (30-s Wingate test) responses to an eleven-day altitude training camp in adolescent runners. To facilitate an improved understanding of the effects of the training camp on running performance, we also sought to explore whether the training camp modulated relationships between running velocity and blood lactate levels, a significant indicator of running performance [20,21,22]. We hypothesized that an appropriately structured, short-duration (i.e., 11-day) altitude training camp would improve aerobic and anaerobic power as well as running velocity at a given blood lactate level.

## 2. Materials and Methods

### 2.1. Subjects

Twenty three middle- and long-distance runners (14–18 yrs) participated in our study. They competed in distances ranging from 800–3000 m. Twelve individuals were randomized (by the r-and-between function in Excel) to the altitude group, which took part in an eleven-day training camp. The altitude group consisted of five females (16.6 ± 0.8 yrs, 60.0 ± 3.4 kg, 168.8 ± 2.5 cm, and 49.0 ± 3.3 mL·kg^−1^·min^−1^) and seven males (16.4 ± 1.7 yrs, 63.7 ± 11.3 kg, 179.6 ± 8.2 cm, and 63.1 ± 4.0 mL·kg^−1^·min^−1^). The control group consisted of eleven members, but three did not complete the research, one was injured and two fell ill with respiratory disease. Four females (17.0 ± 0.6 yrs, 57.8 ± 5.7 kg, 168.0 ± 4.9 cm, and 50.5 ± 2.1 mL·kg^−1^·min^−1^) and four males (16.2 ± 1.0 yrs, 54.7 ± 7.1 kg, 172.0 ± 3.5 cm, and VO_2_max 62.3 ± 1.9 mL·kg^−1^·min^−1^) completed the research program in the control group. All subjects reported having participated in structured run training (6–10×/wk for ~60–90 min per session) for at least two years.

### 2.2. Measures

To facilitate an improved understanding of the effects of short-duration altitude training on aerobic and anaerobic performance indicators in adolescent individuals, we assessed VO_2_max and 30-s Wingate performance in 14–18 yr old middle- and long-distance runners before and after an eleven-day altitude (1850 m above sea level) training camp. Running velocity at various blood lactate levels (2, 4, 6, and 9 mmol·L^−1^) were also measured before and after altitude exposure. In the altitude and control groups alike, we monitored morning heart rate (HR) values for evaluation of training load and physiological adaptation.

### 2.3. Design and Procedures

Runners took part in an eleven-day training camp in Livigno, Italy, a territory in the province of Sondrio at an altitude of ~1850 m above sea level. Training plans of the altitude and control groups were matched for intensity and duration and based on relative heart rate values. The runners in the control group trained in the place of their residence, i.e., at the altitude of ~400 m above sea level. These runners experienced the same training load in terms of intensity (determined by heart rate values) and volume as the members of the altitude group. Thus, the altitude training in the altitude group was the only difference between the groups over the study duration. The training block consisted mostly of low-intensity running below aerobic threshold (AeT) with two higher intensity anaerobic threshold (Ant) [23] training sessions. Over the first three training days, all subjects preformed strictly low intensity training so as to provide the altitude group with a chance to acclimatize to their new environment. Over the ensuing days, the load was gradually increased. In the eleven-days of training, the runners completed ~100 kilometers of running. We used morning heart rate values to evaluate acclimatization to the altitude exposure and training load. Heart rate (HR) was recorded using a Garmin Fenix 3 device (Garmin, Olathe, KS, USA) immediately post-waking and in the seated position. HR values were recorded 10 days before the camp, during the camp, and for 10 days after returning from the camp. Training after returning from training camp was similar in both groups. On the first day after training camp was a day off. The second and third days were low-intensity running below AeT. After this, participants resumed their normal pre-camp training routine. Moreover, all subjects performed a lactate field test one day before the maximal oxygen consumption tests.

Changes in aerobic and anaerobic power were assessed by laboratory testing. Cardiorespiratory fitness (VO_2_max) was assessed via indirect calorimetry (Metalyzer B3, Cortex, Leipzig, Germany) on a cycle ergometer. The following parameters were evaluated at the moment of subjective exhaustion: absolute and relative VO_2_max, tidal volume (V_T_), minute respiratory volume (VE), respiratory frequency (BF), relative work rate (WR) at VO_2_max (WR·kg^−1^), and heart rate at VO_2_max (HR VO_2_max). The tests were carried out three days before leaving for the camp and nine days after returning. In both cases, the same testing protocol was utilized. The test began with a 2-minute warm-up at a load 25 W, after which the load was subsequently adjusted from 2 W·kg^−1^ until volitional exhaustion. Total test duration was between 7–9 min. Criteria used to verify attainment of VO_2_max included: respiratory exchange ratio (RER) > 1.1, rating of perceived exertion (RPE) > 17, achievement of plateau in VO_2_, and achievement of 90% age-predicted maximal heart rate [24]. 

Anaerobic performance was characterized by means of the 30 s Wingate test [18]. The test is used to determine the maximum relative power in 1 s, 5 s, and 30 s. Wingate testing was conducted on an Excalibur sport bicycle ergometer (Lode, Groningen, The Netherlands), 30 min before indirect calorimetry. A five-minute warm-up on the bicycle was followed by a 30 s all-out test (30 s of riding at maximum pedaling frequency, with a load equivalent to 7.5% body mass).

Running speeds at various blood lactate concentrations (2, 4, 6, and 9 mmol·L^−1^) were calculated from a field test (4 × 1600 m on a track), with each repeat performed at a faster velocity. Following each 1600-m interval, participants rested for two minutes while blood lactate was assessed via fingerstick using a Lactate Scout+ (EKF Diagnostics, Cardiff, UK). Blood lactate values at various running velocities were processed by Winlactat (Mesics, Munster, Germany) and an exponential function was used to generate a blood lactate curve, which provided running speeds at a range of blood lactate concentrations of 2, 4, 6, and 9 mmol·L^−1^ [1,25]. Blood lactate measurements were made at an elevation of ~400 m above sea level, two days before leaving for the camp and eight days after return.

### 2.4. Statistical Analysis

Data were represented as the average and the standard deviation (SD), unless otherwise indicated. To verify the normality of the data for both groups (altitude and control), we used the Shapiro–Wilk test, which showed that these data differed significantly from the normal distribution. This led us to use nonparametric methods for further statistical processing. Between group changes were analyzed using the non-parametric Wilcoxon two-sample test and effect sizes (Cohen’s d) were used to provide insight into the magnitude of the differences, where d > 0.8 expresses large effect, a value of 0.5–0.8 expresses an intermediate effect, a value of 0.2–0.5 expresses a small effect, and d < 0.2 signifies a slight effect [26]. Ten-day heart rate value means before, during, and after training camp were calculated using a Wilcoxon matched pairs test. Significance was set at the *p* < 0.05 level. Data processing was performed in Statistica 12 (Dell, Round Rock, TX, USA) and Microsoft Excel (Oregon, WA, USA).

### 2.5. Ethical Approval

All participants, or parents in case of minors, signed an informed consent form. The research was carried out with consent of the Ethics Committee, Faculty of Education, University of South Bohemia, Ref. No.: 001/2018. All procedures performed in the study were in accordance with the ethical standards of the institutional research committee and with the Helsinki declaration.

## 3. Results

Figure 1 provides heart rate values in the altitude group ten days before, during, and after camp. The average post-waking supine HR increased considerably (*p* = 0.0047, d = 0.545) during the 10 days of the training camp in comparison with before 10 days before the training camp, a finding that appeared to be driven by initial (day 1–3) elevations in heart rate. On the ninth day of the camp, the morning heart rate reached typical values. After the return from training camp, average post-waking supine HR decreased considerably (*p* = 0.0029, d = 0.616) in comparison with before the training camp. After the return from training camp, average post-waking supine HR decreased considerably (*p* = 0.0022, d = 1.205) in comparison with the training camp. In the control group, there were no significant changes in HR (Figure 2), *p* = 0.484, d = 0.019 was in the change before camp and during camp, during camp and after camp, *p* = 0.263, d = 0.085, and before camp and after camp, *p* = 0.400, d = 0.064.

Results from maximal exercise testing are provided in Table 1. RER values > 1.10 provide evidence that subjects provided maximal effort [24]. After the return from the training camp, maximum oxygen significantly increased (*p* < 0.01; d = 0.92) in the altitude versus the control group, as well as the peak relative work rate (+9.6%). Significant (*p* < 0.05) improvements in the altitude training group compared with the control group were also observed in minute ventilation (+5.2%) and peak heart rate (+1.4%).

Anaerobic testing results are provided in Table 2. No statistically significant (*p* > 0.05) increases were observed in the altitude versus control group. However, moderate effect sizes were observed for changes in relative peak power (d = 0.47) and relative 5-s peak power (d = 0.59) in the altitude group.

Table 3 provides theoretical running velocities computed at AeT (2 mmol·L^−1^), AnT (4 mmol·L^−1^), 6, and 9 mmol·L^−1^, measured before the camp and after the return. Altitude training camp appeared to positively benefit (*p* < 0.01) running velocity at all blood lactate levels when compared with the control condition.

## 4. Discussion

Short-duration (i.e., 11 days) altitude training significantly improved VO_2_max [8,11,27,28] and velocity at a given blood lactate concentration in trained adolescent runners [11,17,28]. Under optimal load, submaximal parameters (lactate thresholds) can be improved faster than maximum parameters, which was also reflected in our research [10,15,17,29]. These findings mirror what may be expected in their adult counterparts, suggesting similar adaptation of cardiorespiratory fitness from short-duration altitude exposure in trained adolescents and adults. Importantly, the trained adolescent runners from the perspective of individual acclimatization well mastered the planned training at higher altitudes and they appeared to tolerate the training well (see Figure 1). Altitude training is more demanding and requires a reduction in training intensity. Failure to do so may also reduce some fitness parameters [30]. In some cases, a reduction in VO_2_max may occur. [17]. These contrasting findings may be explained by the mastered or not initial three-day adaptation period used in the current study that allowed the runners to acclimate to the training environment.

We checked whether altitude training may serve as a stimulus to enhance anaerobic performance, reflected by an improvement on the Wingate test. Contrary to our hypothesis, 30 s anaerobic power was not affected by the altitude training camp. This finding is in agreement with previous research demonstrating a lack of 30 s Wingate power enhancement in eight young (~20 yrs) team sports players following 4-weeks of simulated altitude (~2750 m) training [31]. More specific training regimens featuring high intensity intervals with sufficient rest (e.g., 2 min+) may be needed if improvements in anaerobic power are desired.

Our use of a cycle ergometer, rather than treadmill for maximal exercise testing is an undeniable limitation of the present study, as runners would likely achieve a superior VO_2_max value on a treadmill. [32,33,34]. Due to the nature of our population (i.e., developing adolescents), we felt that a cycle ergometer was more appropriate to reduce the risk of injury. Despite this limitation, we still observed a 13.6% improvement in VO_2_max following the return from the training camp. In trained runners, this magnitude of change is quite robust over such a short time period, considering 3–6 months of exercise training may only confer improvements of ~20% [35]. In conjunction with greater running velocity at various blood lactate concentrations, this improvement in VO_2_max portends to the likelihood of significant race performance improvements in these young athletes. Indeed, running velocity-blood lactate relationships are an integral component of running performance [11,36,37,38]. Continual tracking of these performance variables following the return to sea level would provide insight as to the persistence of these observed adaptations, thus helping to inform training strategies for athletes and coaches looking to employ altitude training as a means to improve race performance. Moreover, it would be interesting to examine the effects of longer-duration altitude training on these performance variables.

Limitations of our study include the use of a cycle ergometer instead of a treadmill, which is more specific for the population, and the relatively small number of participants. However, due to cost and time commitments, it is difficult to recruit a large sample for such a study. Strengths of our study include a control group, which was neglected in many previous altitude training studies, as well as a thorough testing battery including aerobic and anaerobic performance indicators.

## 5. Conclusions

An eleven-day altitude training camp at an altitude of ~1850 m above sea level proved highly effective at enhancing cardiorespiratory fitness and running velocity at various blood lactate levels, but not anaerobic performance, in trained adolescent runners. These findings are comparable to what may be observed in adult counterparts, and highlight the value of appropriately structured short-duration altitude training for improving fitness level and likely race performance at sea level in this population.

## Figures and Tables

**Figure 1 medicina-56-00184-f001:**
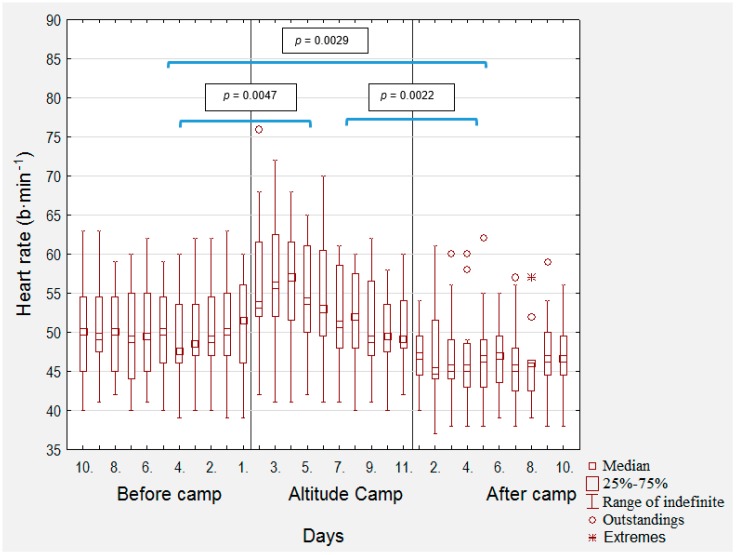
Morning heart rate 10 days before the camp, during the camp, and 10 days after the camp in the altitude group, and statistical significance of changes between particular parts of training camp. Legend: b10–b1: days 10.–1. before camp, c1–c11: 1.–11. camp days, a1–a10: days 1.–10. after camp.

**Figure 2 medicina-56-00184-f002:**
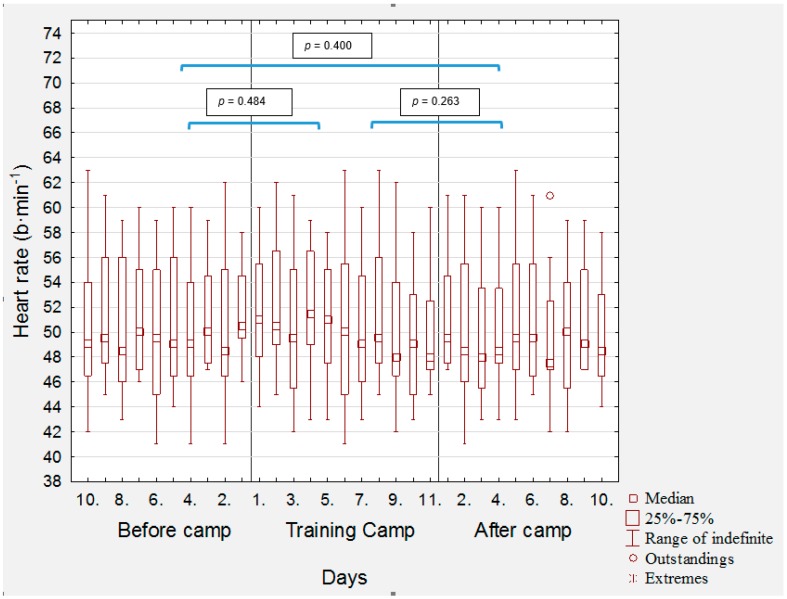
Morning heart rate in the control group 10 days before the camp, during the camp, and 10 days after the camp, and statistical significance of changes between particular parts of training camp. Legend: b10–b1: days 10.–1. before camp, c1–c11: 1.–11. camp days, a1–a10: days 1.–10. after camp.

**Table 1 medicina-56-00184-t001:** Cardiorespiratory fitness testing results before and after the eleven day altitude training camp of the altitude group and control group (training at 400 m above sea level).

VO_2_max (mL·min^−1^·kg^−1^)	Before	After	% Change	Cohen’s d	Statistical Significance
altitude group	57.3 ± 7.9	64.9 ± 8.8	13.55	0.92	*p* < 0.01
control group	56.4 ± 6.2	56.8 ± 6.4	0.66	0.06	
VT (L)	before	after	% change	Cohen’s d	statist. significance
altitude group	2.519 ± 0.522	2.642 ± 0.537	5.16	0.23	*p* > 0.05
control group	2.309 ± 0.575	2.300 ± 0.558	−0.24	−0.02	
VE (L·min^−1^)	before	after	% change	Cohen’s d	statist. significance
altitude group	134.6 ± 37.0	148.1 ± 31.1	12.34	0.41	*p* < 0.05
control group	124.7 ± 18.5	124.9 ± 18.9	0.16	0.01	
BF (b·min^−1^)	before	after	% change	Cohen’s d	statist. significance
altitude group	54.1 ± 11.9	57.2 ± 10.6	7.47	0.27	*p* > 0.05
control group	56.4 ± 13.5	56.5 ± 12.2	0.83	0.01	
RER	before	after	% change	Cohen’s d	statist. significance
altitude group	1.163 ± 0.053	1.130 ± 0.029	−2.74	0.79	*p* > 0.05
control group	1.125 ± 0.037	1.125 ± 0.023	0.06	0.00	
SF VO_2_max (beat·min^−1^)	before	after	% change	Cohen’s d	statist. significance
altitude group	183.8 ± 7.6	186.2 ± 8.1	1.35	0.31	*p* < 0.01
control group	189.0 ± 10.6	188.9 ± 9.9	−0.04	0.02	
WR (W·kg^−1^)	before	after	% change	Cohen’s d	statist. significance
altitude group	4.57 ± 0.69	4.95 ± 0.52	9.57	0.63	*p* < 0.01
control group	4.27 ± 1.04	4.26 ± 1.05	−0.37	−0.01	

**Table 2 medicina-56-00184-t002:** Anaerobic 30-s Wingate power testing results before and after the eleven-day altitude training camp of the altitude group and control group (training at 400 m above sea level).

Relative Peak 30 s Power (W·kg^−1^)	Before	After	% Change	Cohen’s d	Statistical Significance
altitude group	8.091 ± 0.877	8.525 ± 0.977	5.12	0.47	*p* > 0.05
control group	7.976 ± 0.900	8.016 ± 1.067	0.86	0.06	
relative peak power (W·kg^−1^)	before	after	% change	Cohen’s d	statist. significance
altitude group	11.466 ± 1.587	12.547 ± 2.027	7.39	0.39	*p* > 0.05
control group	11.671 ± 1.601	11.714 ± 1.646	0.19	0.03	
relative peak 5 s power (W·kg^−1^)	before	after	% change	Cohen’s d	statist. significance
altitude group	9.474 ± 1.466	10.041 ± 1.412	9.38	0.59	*p* > 0.05
control group	9.666 ± 1.393	9.750 ± 1.419	0.86	0.06	
revs per 30 s	before	after	% change	Cohen’s d	statist. significance
altitude group	59.7 ± 4.9	62.2 ± 5.3	4.61	0.50	*p* > 0.05
control group	60.0 ± 3.6	60.1 ± 3.3	0.25	0.04	

**Table 3 medicina-56-00184-t003:** Estimated running speeds at 2, 4, 6, and 9 mmol·L^−1^ and SD, and changes before and after the camp determined from field testing by Winlactat software.

2 mmol·L^−1^	Running Speed before (km·h^−1^)	Running Speed after (km·h−^1^)	% Change	Cohen’s d	Statistical Significance
altitude group	12.04 ± 1.58	12.97 ± 1.65	7.93	0.58	*p* < 0.01
control group	12.91 ± 1.80	13.00 ± 1.82	0.75	0.05	
**4 mmol·L^−1^**	running speed before (km·h^−1^)	running speed after (km·h^−1^)	% change	Cohen’s d	statist. significance
altitude group	13.73 ± 1.77	14.84 ± 1.58	8.47	0.66	*p* < 0.01
control group	14.00 ± 2.09	14.10 ± 2.14	0.74	0.05	
**6 mmol·L^−1^**	running speed before (km·h^−1^)	running speed after (km·h^−1^)	% change	Cohen’s d	statist. significance
altitude group	14.57 ± 1.51	15.93 ± 1.57	9.59	0.88	*p* < 0.01
control group	14.55 ± 1.95	14.65 ± 2.01	0.63	0.05	
**9 mmol·L^−1^**	running speed before (km·h^−11^)	running speed after (km·h^−1)^	% change	Cohen’s d	statist. significance
altitude group	15.92 ± 1.62	16.82 ± 1.53	5.87	0.57	*p* < 0.01
control group	15.60± 1.84	15.76 ± 1.85	1.03	0.09	

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
