# Peer review of "Effect of an Eleven-Day Altitude Training Program on Aerobic and Anaerobic Performance in Adolescent Runners"

_medicina, 2020, doi:10.3390/medicina56040184_

Round 1

Reviewer 1 Report

I would like to acknowledge the authors’ effort and time put into preparing this paper. The paper interested topic. However, in my opinion, there are some major limitation that need to be considered and revised before publishing the results of this study.

Since the hypothesis of the study is “that an appropriately structured, short-duration (i.e., 11 day) altitude training camp would improve aerobic and anaerobic power as well as running velocity at a given blood lactate level” the key issues are:

  • How the randomization was applied? Why, from the group of 20 athletes, 14 were in the enrolled to the experimental, and only 8 runners were enrolled to the control group
  • Since the laboratory tests were performed three days before the camp and nine days after returning from the camp, can you provide detailed information in terms of monitoring the training intensity and volume through nine days; from the last day of the camp to the day of laboratory tests?
  • What kind of statistical tests were used to determine if a data set was well-modeled by a normal distribution; it is rather unique to have normal distribution if there is only data from eight participants in control group
  • It is written in lines 127-128 that “Ten-day heart rate values means before, during, and after training camp were calculated and compared using a one-way ANOVA.” On the other hand, the Figure 1 presents data of heart rate expressed as median and ranges of indefinite. It should be clarified, what type of statistical analysis was applied, whether parametric or non-parametric.
  • Why the data presented in Figure 1 are not presented separately for the experimental and control group
  • In table 1 VT should be corrected to Vt and L should be corrected to l
  • In table 3: can you please specify the unit of measure: km.hod-1 – what do you mean?
  • The discussion could be better elaborated citing more relevant publications. There is only 5 research cited in this section, that seems limited number regarding the number of publications analyzing the effect of altitude training programs in runners. E.g. no relevant explanation for running speed improvement is discussed.

All my remarks and suggestions mentioned above are only motivated by the will to improve the quality of the paper. However, the shortcomings of the paper do not allow me to accept the manuscript in the current version. ​

Author Response

Response to Reviewer 1 Comments

  • Point 1: How the randomization was applied? Why, from the group of 20 athletes, 14 were in the enrolled to the experimental, and only 8 runners were enrolled to the control group

Response 1: Line 71: In the begining, twenty three middle- and long-distance runners (14-18 yrs) participated in our study. Twelve individuals were randomized (by the r-and-between function in Excel) to the altitude group, which took part in an eleven-day training camp. The control group consisted of eleven members, but three did not complete the research, one was injured and two fell ill with respiratory disease.

  • Point 2: Since the laboratory tests were performed three days before the camp and nine days after returning from the camp, can you provide detailed information in terms of monitoring the training intensity and volume through nine days; from the last day of the camp to the day of laboratory tests?

Response 2: Line 101: Training after returning from training camp was similar in both groups. On the first day after training camp was a day off. The second and third days were low-intensity running below AeT. After this, participants resumed their normal pre-camp training routine. Moreover, all subjects performed a lactate field test one-day before the maximal oxygen consumption tests.

 Point 3: What kind of statistical tests were used to determine if a data set was well-modeled by a normal distribution; it is rather unique to have normal distribution if there is only data from eight participants in control group

Response 3: Line 131: We apologize for this oversight and appreciate the reviewer bringing it to our attention. Indeed, data were not normally distributed and thus substantial revisions were required. Line 132: To verify the normality of the data for both groups (altitude and control), we used the Shapiro - Wilk test, which showed that these data differ significantly from the normal distribution. This led us to use nonparametric methods for further statistical processing. Between group changes were analysed using the non-parametric Wilcoxon two-sample test and effect sizes (Cohen’s d) were used to provide insight into the magnitude of the differences, where d > 0.8 expresses large effect, a value of 0.5-0.8 expresses an intermediate effect, a value of 0.2-0.5 expresses a small effect, and d < 0.2 signifies a slight effect [24]. Ten-day heart rate values means before, during, and after training camp were calculated using a Wilcoxon matched pairs test. Significance was set at the P < 0.05 level. Data processing was performed in Statistica 12 and Microsoft Excel.

 Point 4: It is written in lines 127-128 that “Ten-day heart rate values means before, during, and after training camp were calculated and compared using a one-way ANOVA.” On the other hand, the Figure 1 presents data of heart rate expressed as median and ranges of indefinite. It should be clarified, what type of statistical analysis was applied, whether parametric or non-parametric.

Response 4: Due to the non-normality of our data we have revised our statistical procedures. Line 137: Ten-day heart rate value means before, during, and after training camp were calculated using a Wilcoxon matched pairs test.

 Point 5: Why the data presented in Figure 1 are not presented separately for the experimental and control group

Response 5: 161: We have added a new figure (Figure 2) illustraing heart rate values in the control group.

 Point 6: In table 1 VT should be corrected to Vtand L should be corrected to l

Response 6: Line 173: Corrected in the table

  • Point 7: In table 3: can you please specify the unit of measure: km.hod-1 – what do you mean?

Response 7: Line 190: It was mistake, corrected

 Point 8: The discussion could be better elaborated citing more relevant publications. There is only 5 research cited in this section, that seems limited number regarding the number of publications analyzing the effect of altitude training programs in runners. E.g. no relevant explanation for running speed improvement is discussed.

Response 8: Line 191: Corrected. Under optimal load, submaximal parameters (lactate thresholds) can be improved faster than maximum parameters, which was also reflected in our research (Sources in lines: 256, 261, 268, 270, 278, 303)

Reviewer 2 Report

Overall, this manuscript describes a scientifically-sound study of interest to researchers involved in cardiovascular health and running. Most of my comments are minor and I am confident the authors can address them fully. Specific comments are as follows: Abstract: Line 18 - Adolescent is repeated, please remove. Line 23 - You use VO2peak in your abstract, but VO2max in the body of the manuscript. I would recommend choosing one and using it throughout. VO2 max may be more readily recognized by readers. Line 28 - If possible, please provide the exact p-value for changes in anaerobic power. Introduction: You do a nice job of summarizing the literature. However, I do not feel that you have built a sufficient case for why studying these adaptations in adolescent runners is needed, beyond the fact that this population has not yet been studied in this way. Thinking of the big picture, why do we need to know how adolescent runners adapt to training at altitude? Is it simply improved performance/competitive results? Or are there long-term health benefits? Or other benefits to younger athletes training at altitude that may differ from adult counterparts? Methods: Your subjects section is very clear and specific. Nice job. Measures: I would recommend revising to remove the word "compared", as that is part of your statistical analysis. Revise this section to purely describe what was measured and when it was measured. Design & Procedures: This is also clear and specific. I would recommend using either "altitude group" or "experimental group" throughout to make it easier for your reader to understand. You switch back and forth a few times beginning in this section. Personally, I think altitude group is easier to follow, but as long as it is consistent throughout, I would be fine with either. Addition of a schematic of the training program may be beneficial as well. Line 100: Comment here, or elaborate further in the discussion, on why a cycle ergometer was chosen. Previous research generally shows that VO2max is lower on a cycle ergometer than a treadmill, particularly in non-cyclists, which should be OK since you are more concerned with change values. I would recommend, however, discussing or commenting on the reliability of cycle ergometers compared to treadmill tests. They are generally both reliable but the cycle tests do tend to have more variability. It would be important to discuss you change values within the context of variability on the cycle ergometer and also in the context of MDC/MCID if available. Statistical analysis: I am curious as to why you did a one-way ANOVA for heart rate changes. Were you not interested in the potential interaction between time and group? Results: Line 142: Please provide the p-value associated with this result. Line 145: Add statistical findings to the caption for Figure 1. Line 153: So that this table could stand alone, please clarify that the control group was not at altitude in the caption for table 1. Discussion: Line 186: Given that you describe previous research that altitude training does not improve Wingate power, why was your hypothesis that altitude training would improve power? Line 199: Similar to my comment above, please elaborate on potential limitations of using the cycle ergometer.

Author Response

Response to Reviewer 2 Comments

Point 1:  Abstract: Line 18 - Adolescent is repeated, please remove.

Response 1: Line 18: Corrected

Point 2:  Line 22 - You use VO2peak in your abstract, but VO2max in the body of the manuscript. I would recommend choosing one and using it throughout. VO2 max may be more readily recognized by readers.

Response 2: We agree and chose VO2max and updated our manuscript throughout

Point 3: Line 27 - If possible, please provide the exact p-value for changes in anaerobic power.

Response 3: Because we used the non-parametric Wilcoxon two-sample test, p-value was not calculated in this case. We calculated Min (U1,U2) for determine the siginificance level.

Point 4: Introduction: You do a nice job of summarizing the literature. However, I do not feel that you have built a sufficient case for why studying these adaptations in adolescent runners is needed, beyond the fact that this population has not yet been studied in this way. Thinking of the big picture, why do we need to know how adolescent runners adapt to training at altitude? Is it simply improved performance/competitive results? Or are there long-term health benefits? Or other benefits to younger athletes training at altitude that may differ from adult counterparts?

Response 4: Line 53-57: Since there is no doubt that training at higher altitudes can benefit peak performance in adult runners [5,7,10,11], and because acclimatization and training become more effective with every stay at higher altitude [19], it is interesting to see how young runners respond to these conditions. 

Point 5: Methods: Your subjects section is very clear and specific. Nice job.

Response 5: Thank you

Point 6: Measures: I would recommend revising to remove the word "compared", as that is part of your statistical analysis. Revise this section to purely describe what was measured and when it was measured.

Response 6: Lines 78-84: corrected

Point 7: Design & Procedures: This is also clear and specific. I would recommend using either "altitude group" or "experimental group" throughout to make it easier for your reader to understand. You switch back and forth a few times beginning in this section. Personally, I think altitude group is easier to follow, but as long as it is consistent throughout, I would be fine with either. Addition of a schematic of the training program may be beneficial as well.

Response 7: We agree with the reviewer and have revised to make consistent throughout by only using altitude group.

Point 8: Line 100: Comment here, or elaborate further in the discussion, on why a cycle ergometer was chosen. Previous research generally shows that VO2max is lower on a cycle ergometer than a treadmill, particularly in non-cyclists, which should be OK since you are more concerned with change values. I would recommend, however, discussing or commenting on the reliability of cycle ergometers compared to treadmill tests. They are generally both reliable but the cycle tests do tend to have more variability.

Response 8: Line 211: Indeed, use of a cycle ergoemter rather than a treadmill is a limitation of the present study. However, maximal exercise testing on a cycle ergometer was strategically chosen to reduce the risk of injury in this adolescent populations. Nonetheless, we have mentioned this as a limitation in our discussion section

Point 9: It would be important to discuss you change values within the context of variability on the cycle ergometer and also in the context of MDC/MCID if available.

Response 9: To provide context to the observed changes we have cited previous literature showing changes in VO2max following a long-duraiton training program (lines 276-277).

Point 10: Statistical analysis: I am curious as to why you did a one-way ANOVA for heart rate changes. Were you not interested in the potential interaction between time and group?

Response 10: Line 130: We corrected. To verify the normality of the data for both groups (altitude and control), we used the Shapiro - Wilk test, which showed that these data differ significantly from the normal distribution. This led us to use nonparametric methods for further statistical processing. Between group changes were analysed using the non-parametric Wilcoxon two-sample test.

Point 11: Results: Line 142: Please provide the p-value associated with this result. Line 145: Add statistical findings to the caption for Figure 1.

Response 12: Line 156: It is showed in figure.

Point 12: Line 153: So that this table could stand alone, please clarify that the control group was not at altitude in the caption for table 1.

Response 12: Line 173: Corrected: control group (training at 400 m above see level)

Point 13: Discussion: Line 186: Given that you describe previous research that altitude training does not improve Wingate power, why was your hypothesis that altitude training would improve power?

Response 13: We investigated Wingate power for a comprehensive evaluation of changes in training.

Point 14: Line 199: Similar to my comment above, please elaborate on potential limitations of using the cycle ergometer. 

Response 14: Line 211: Added in discussion